# MulCPred: Learning Multi-Modal Concepts for Explainable Pedestrian Action Prediction

**DOI:** 10.3390/s24206742

**Published:** 2024-10-20

**Authors:** Yan Feng, Alexander Carballo, Keisuke Fujii, Robin Karlsson, Ming Ding, Kazuya Takeda

**Affiliations:** 1Graduate School of Informatics, Nagoya University, Furo-cho, Chikusa-ku, Nagoya 464-8601, Japan; 2Institutes of Innovation for Future Society, Nagoya University, Furo-cho, Chikusa-ku, Nagoya 464-8601, Japan; 3Graduate School of Engineering, Gifu University, 1-1 Yanagido, Gifu City 501-1193, Japan; 4Tier IV Inc., Nagoya University Open Innovation Center, 1-3, Mei-eki 1-Chome, Nakamura-ward, Nagoya 450-6610, Japan

**Keywords:** pedestrian action prediction, computer vision, neural networks, multi-modal learning, explainable AI, autonomous driving

## Abstract

Pedestrian action prediction is crucial for many applications such as autonomous driving. However, state-of-the-art methods lack the explainability needed for trustworthy predictions. In this paper, a novel framework called MulCPred is proposed that explains its predictions based on multi-modal concepts represented by training samples. Previous concept-based methods have limitations, including the following: (1) they cannot be directly applied to multi-modal cases; (2) they lack the locality needed to attend to details in the inputs; (3) they are susceptible to mode collapse. These limitations are tackled accordingly through the following approaches: (1) a linear aggregator to integrate the activation results of the concepts into predictions, which associates concepts of different modalities and provides ante hoc explanations of the relevance between the concepts and the predictions; (2) a channel-wise recalibration module that attends to local spatiotemporal regions, which enables the concepts with locality; (3) a feature regularization loss that encourages the concepts to learn diverse patterns. MulCPred is evaluated on multiple datasets and tasks. Both qualitative and quantitative results demonstrate that MulCPred is promising in improving the explainability of pedestrian action prediction without obvious performance degradation. Moreover, by removing unrecognizable concepts, MulCPred shows improved cross-dataset prediction performance, suggesting its potential for further generalization.

## 1. Introduction

Autonomous vehicles should be able to understand the intentions of other road users to navigate safely and efficiently in urban traffic environments, especially those of pedestrians [1,2,3]. Pedestrian behaviors often exhibit diversity and randomness, which makes predicting their future actions particularly challenging. Recent years have seen significant advances in pedestrian behavior prediction, including trajectory [4,5,6,7,8,9,10] or action category predictions [11,12,13,14,15,16,17,18]. Pioneering works have exploited uni-modal features such as historical trajectories [6,7] and poses [14]. Recent works have used more sophisticated models and multiple modalities of inputs [5,11,18]. Typically, these “multiple modalities” involve different aspects of information manually selected from sensor data, such as skeletons, appearance, and trajectories. Such manual disentanglement of raw inputs has so far been effective in exploiting limited training data and has shown improvements in prediction accuracy. Despite the improvement in prediction performance, recent learning-based models still face the lack of explainability in decision-making, due to the intrinsic black-box characteristic of deep neural networks, which presents difficulties for users to understand the working principle and for the developers to make further improvements.

Existing works in explainable artificial intelligence (XAI) [19,20] have made significant progress in fundamental tasks such as image recognition and speech recognition, among which concept-based models [21,22,23,24,25] as a category of self-explaining methods received wide attention. The basic idea of concept-based models is to decompose the information extracted by a deep backbone to a small number of basis concepts. Consider the target pedestrian in Figure 1: in order to understand the road crossing action, our experiences make us observe elements such as the road, the sidewalk, the crosswalk, the relative position of the pedestrian with cars and other pedestrians, the trajectory of the pedestrian, the buildings, etc. Some may have higher relevance for the action (e.g., the crosswalk) than others. In our work, we refer to all these elements as “concepts”; our approach can extract these elements from the image sequence, weigh their relevance to the action, and combine them in order to achieve accurate action prediction, which is interpretable and supported by the evidence provided by the concepts. Each concept is designed to represent a specific pattern in the input, allowing samples with high activation values for the concept to serve as explanations. While these methods have proven effective in basic tasks like image recognition, they face several challenges when applied to a broader range of applications. First of all, existing methods only considered uni-modal cases, such as images and speech signals and, thus, lack the means to integrate and compare the information from different modalities. Secondly, these methods only achieve sample-level explainability, meaning the concepts can only be represented by samples, but not detailed elements in the samples, which is problematic in our tasks since urban traffic scenarios contain rich information and the inter-class difference of pedestrian behaviors is usually weak. Thirdly, existing methods sometimes suffer from mode collapse, which would seriously degrade the explainability.

In this paper, we present MulCPred, a self-explaining framework for pedestrian action prediction with multi-modal inputs. Figure 1 illustrates the inference process of MulCPred. The inputs are projected to a set of concepts. Each concept corresponds to an activation score that represents the similarity between the input and the concept. These activation scores are calculated by a channel-wise recalibration module that learns a channel-wise distribution to select different components in the features. We use a linear aggregator to explicitly integrate activation scores of concepts from multiple modalities, which enables the model with better ante hoc explainability by revealing the relevance between the concepts and the labels.

The contributions of this work are as follows:We present a self-explaining framework for pedestrian behavior prediction for multi-modal inputs. A linear layer is used as an aggregator to combine the activation results of concepts from different modalities. The linear layer also provides ante hoc explanations for the relevance between the concepts and the predictions, which further improves the overall explainability of the framework.We propose a channel-wise recalibration module for each concept to attend to local spatiotemporal regions to improve the locality of the concepts.We propose a feature regularization loss term to promote different concepts to learn diverse content. The visualization of the concepts shows that the feature regularization loss term effectively prevents the concepts from collapsing to limited patterns.The proposed framework achieves competitive performance on two tasks: pedestrian crossing prediction and atomic action prediction. Moreover, we show that by removing concepts that are not understandable, the cross-dataset prediction performance of MulCPred is also improved, meaning we can manually improve the generalizability by manipulating the learned concepts.We extend the most relevant first (MoRF) curve [26], a perturbation-based faithfulness metric, to apply to negative relevance. The extended MoRF curve demonstrates the faithfulness of explanations generated by the model.We have made the code available at https://github.com/Equinoxxxxx/MulCPred_code (accessed on 20 May 2024).

## 2. Related Works

### 2.1. Pedestrian Behavior Prediction

Existing works mainly focus on two forms of pedestrian behaviors: (1) trajectory, which is the sequence of coordinates on the ground plane or the image plane, and (2) action categories that the pedestrian will conduct in future frames.

**Trajectory prediction.** Predicting behaviors of pedestrians in the form of trajectories has been extensively studied, among which there are two main application scenarios: the surveillance view [6,7,8] and the ego-centric view [4,11,13,27]. In surveillance scenarios, the fixed view enables methods to leverage pedestrians’ kinetic features, such as historical trajectories, as well as inferring interactions between pedestrians. On the other hand, the perspective of view and ego-motion of onboard cameras bring difficulties to utilizing interaction information. Therefore, ego-centric methods usually use multi-modal inputs, including ego-motion and the pedestrians’ appearance information [5,11]. Some also use estimated intention or action categories [27,28] of the pedestrians to enhance the prediction. Experiments in these works show that accurate estimation of the pedestrian’s intention or action can also improve the prediction of trajectories.

**Action prediction**. A number of pedestrian action prediction methods focus on the crossing behaviors, i.e., whether the observed pedestrian will cross in front of the ego vehicle. Early works exploited uni-modal features such as poses [14] and fine-grained action categories of upper and lower bodies [13]. These methods suffer from low prediction accuracy due to their relatively weak modeling capability. During the last decade, the performance of action prediction has been significantly improved as model complexity increased. Recently proposed works are multi-modal methods that jointly encode the inputs from multiple modalities, such as original images from onboard cameras, skeletons of pedestrians, and ego-motion of the vehicle. One common paradigm of these methods is to use separate encoders to generate fixed-length representations of different modalities, and then fuse these representations at a late stage [11,18,27]. Recently there have also been works using more sophisticated fusing strategies. For example, [29] uses a mixed architecture that combines early fusion and late fusion for multiple modalities.

Although these approaches improve performance, the fusion strategies in these approaches (e.g., concatenation, sum, or mixed ones) make it hard to conduct a direct attribution of different components in the inputs and, thus, weaken the explainability.

### 2.2. Explainable Models

Works related to XAI can be roughly categorized into two groups: post hoc methods and ante hoc methods. post hoc methods use additional modules or intermediate results generated during inference to explain the inference itself. Existing post hoc methods include probing-based methods [30,31] and activation map-based methods [32,33,34]. Recently, linguistic expressions as a form of explanation have received extensive attention [35,36,37]. Such methods use language models to generate texts that explain the decision-making process. Despite that natural language is more user-friendly and produces less ambiguity compared with visual modalities, these models still fall within the category of post hoc explaining, since their explanations are sample-specific, meaning such models do not provide prior clues about the predictions to be made.

On the other hand, ante hoc methods, such as linear models and decision trees, are intrinsically interpretable [38], which means explanations of these models do not completely rely on the inference results. One domain that has grown popular in recent years is concept-based methods [21,22,23,24,25,39,40,41]. Such methods learn a set of implicit or explicit concepts and use the similarity between the inputs and these concepts to conduct inference. For instance, ref. [21] proposed a general concept learning framework that is composed of two branches: one for encoding the inputs into a set of concepts, and the other for calculating the relevance weights corresponding to the concepts. Recent works [22,24] have trained concepts in a supervised manner, which ensures that the concepts have diverse and clear meanings. However, this approach increases the workload of annotating and reduces scalability.

Most existing methods concentrate on fundamental tasks, such as image recognition, and do not address multi-modal cases. The learned concepts are at the sample level and lack locality, which can lead to confusion when the sample contains diverse information. We propose a linear aggregator and a feature recalibration module to tackle these limitations. Moreover, mode collapse was observed in the experiments and the literature [40], meaning the concepts learned very few patterns (see Section 4 for details). To curb such trends, we propose a feature regularization loss that encourages the concepts to learn diverse patterns. Note that MulCPred is different from the method in [39] since MulCPred learns the recalibration of the feature extracted by the backbone as the concept, while [39] learns a fixed vector as the concept itself. In such a manner, MulCPred can be multi-modal data, including spatiotemporal data such as videos, or sequential data such as trajectories.

## 3. MulCPred: Multi-Modal Concept-Based Pedestrian Action Prediction

### 3.1. Overview

The overall illustration of MulCPred is shown in Figure 2. The framework consists of two main parts: the multi-modal concept encoders and the linear aggregator. The multi-modal concept encoders first project the input into a set of implicit concepts. Each concept represents a certain pattern of the corresponding data modality. The output of the concept encoders is a set of activation scores, which represent the “similarity” between the input and all the concepts. The activation scores are then integrated by a linear aggregator to predict the confidence of action classes, such that each parameter in the aggregator shows the statistical relationship between a concept and a labeled action.

In the following subsections, we discuss the implementation of our model in detail.

### 3.2. Multi-Modal Concept Encoders

The multi-modal concept encoders first extract fixed-length representations from the input. Then the recalibration modules separate the representations into basic concepts. Let (X,y) be an input-label pair, where $y\in R^{C}$ is a one-hot vector representing an action label out of *C* classes. Each sample $X={\left\{x_{m}\right\}}_{m=1}^{M}$ represents inputs with *M* modalities, where $x_{m}$ can be any form of data, e.g., sequential inputs such as trajectories or spatiotemporal inputs such as videos. For each $x_{m}$, there is a backbone function $f_{m}\left(\cdot \right)$ that extracts feature $f_{m}\left(x_{m}\right)$. The extracted feature is fed into the recalibration module to calculate the activation scores $s_{m}\in R^{N_{m}}$ to a set of concepts. Each element in $s_{m}$ is the activation score corresponding to a certain concept, representing how similar the input is to the concept.

### 3.3. Recalibration Module

The recalibration module is embedded in each modality branch and determines the concepts the model learns. For each modality, the extracted feature $f_{m}\left(x_{m}\right)$ is flattened by a global average pooling layer. Let $r_{m}$ denote the global representation of $f_{m}\left(x_{m}\right)$ after pooling. We first calculate a set of recalibration vectors $P_{m}$ conditioned on $r_{m}$ using an MLP layer:(1)$$r_{m}=\mathrm{GlobalAveragePooling}\left(f_{m}\left(x_{m}\right)\right)$$
(2)$$P_{m}=\mathrm{ReLU}\left(r_{m}W_{m}+b\right)$$
where $r_{m}\in R^{d_{m}}$ is the feature after a global pooling layer, and $P_{m}\in R^{Nd_{m}}$ is the concatenation of ${\left\{p_{m,i}\right\}}_{i=1}^{N}$, with m∈[1,M] corresponding to a certain modality and *N* is the number of concepts for each modality. Each $p_{m,i}\in R^{d_{m}}$ has the same dimension as $r_{m}$, and represents a distribution of different components in $r_{m}$. Finally, the activation score of a certain concept is given by applying dot product between $r_{m}$ and normalized $p_{m,i}$
(3)$$\begin{array}{c} \begin{array}{c} s_{m,i}=\mathrm{softmax}{\left(p_{m,i}\right)}^{⊤}r_{m} \end{array} \end{array}$$
where $s_{m,i}\in R$ is the activation score corresponding to concept i.

Figure 3 shows the visualizing process of concepts for spatiotemporal modalities. Since each recalibration vector $p_{m,i}$ describes a distribution of different components in $r_{m}$, which is the compact form of $f_{m}\left(x_{m}\right)$, the relevance in $p_{m,i}$ can also apply to $f_{m}\left(x_{m}\right)$. Therefore, we can see the spatiotemporal regions that activate a certain concept through the channel-wise multiplication between $f_{m}\left(x_{m}\right)$ and $p_{m,i}$. Such a manner also makes it possible for different concepts can match different patterns even from the same sample.

### 3.4. Aggregator

We use a linear layer as the aggregator to integrate the activation scores of all concepts. Assuming there are N×M concepts equally distributed among all *M* modalities, the activation scores from all *M* modalities are finally integrated by a linear aggregator $g_{W}\left(\cdot \right)$ parameterized by $W_{a}\in R^{NM\times C}$, where *C* denotes the number of predicted classes. Thus, the prediction is given by the following:(4)$$\begin{array}{c} \hat{y}=\mathrm{softmax}\left(\mathrm{ReLU}\left(S\right)W_{a}\right) \end{array}$$
where $S\in R^{NM}$ is the concatenation of activation scores from all modalities. ReLU(·) is used to ensure non-negative activation. Each element at row *i* and volume *j* in $W_{a}$ represents the relevance between concept *i* and class *j*. By explicitly learning the weights, the model’s prediction can be explained by attributing the relevance to each concept and the corresponding activation intensity, which provides ante hoc explainability.

### 3.5. Loss Functions

In the experiments, we discovered that the concepts tend to collapse into few patterns, resulting in the representative samples for most concepts being exactly the same (see Section 4 for details). A similar problem was also reported in [40]. To curb such a trend, we propose a feature regularization loss that contains two terms: the diversity term $L_{div}$ and the contrastive term $L_{cont}$. $L_{div}$ is given by the following:(5)$$\begin{array}{c} L_{div}=\sum_{m=1}^{M}{∥p_{m}p_{m}^{⊤}-I∥}_{2} \end{array}$$
where $I\in R^{N\times N}$ is the identity matrix. This term encourages each row in $p_{m}$ to be different from the rest, meaning that the concepts are encouraged to learn different combinations of channels in the feature $r_{m}$. However, $L_{div}$ alone cannot ensure that the channels in $r_{m}$ represent different patterns. Therefore, we use $L_{cont}$ to promote the diversity of components in $r_{m}$. Assuming that $R_{m}\in R^{B\times d_{m}}$ is a batch of features $r_{m}$ with batch size *B*, we want the columns in $R_{m}$ to be different, which means we want each channel in the feature $r_{m}$ to have a unique batch-wise distribution. Therefore, $L_{cont}$ is given by the following:(6)$$\begin{array}{c} L_{cont}=\sum_{m=1}^{M}{∥R_{m}^{⊤}R_{m}-I∥}_{2} \end{array}$$

The loss of the whole model is composed as follows:(7)$$\begin{array}{c} L=L_{cls}+\lambda_{1}\left(\lambda_{2}L_{cont}+\left(1-\lambda_{2}\right)L_{div}\right) \end{array}$$
where $L_{cls}$ denotes the classification loss. In this paper, we use the cross-entropy loss. $\lambda_{1}$ and $\lambda_{2}$ are the coefficient factors. We set $\lambda_{1}=0.1$ and $\lambda_{2}=0.5$ in practice.

## 4. Experiment

### 4.1. Implementation Details

In the experiments, we use up to five different input modalities.

*Appearance* of the observed pedestrians as sequences of image patches cropped according to the annotated bounding boxes of the pedestrians.*Skeleton* information of the pedestrians is represented in the form of pseudo heatmaps [42], predicted by pre-trained HRNet [43].*Local context* information in the form of sequences of image patches cropped by enlarged bounding boxes around the pedestrians.*Trajectory* information in the form of sequences of four-dimensional coordinates of bounding boxes of the pedestrians.*Ego-motion* of the vehicle, as sequences of the acceleration of the ego vehicle.

Similar to [13], we use an observation length of 16 frames (1.6 s) for all modalities, and predict the action label of the next frame. We use C3D [44] pre-trained on Kinetics 700 [45] as the backbone for the appearance and context modalities. For the skeleton modality, we use PoseC3D [42] pre-trained on UCF101 [46] as the backbone. And for the trajectory and ego-motion modalities, we use randomly initialized LSTMs as the backbone.

During training, we use the Adam optimizer [47] with $\beta_{1}=0.9$, $\beta_{2}=0.999$. λ is 0.01 and η is 1. We use the learning rate of $10^{-5}$ for backbones with pre-trained weights and $10^{-4}$ for other trainable parameters. We use step decay for the learning rate with a step size of 20 and a decay factor of 0.1. The model is trained for 100 epochs with batch size of 8. We set *N*, the number of concepts for each modality, as 10.

### 4.2. Datasets

We evaluated MulCPred on two tasks: pedestrian crossing prediction, and pedestrian atomic action prediction. The crossing prediction task contains both datasets, TITAN and PIE, while the atomic action prediction only contains TITAN.

**TITAN** [48] contains 10 h of 60 FPS driving videos in Tokyo annotated with bounding boxes and action class labels at 10 Hz annotating frequency. The dataset provides multiple independent action sets: communicative actions, transport actions, complex contextual actions, simple contextual actions, and atomic actions. For crossing prediction, we use the simple contextual action set, as it provides two actions specifically involving crossing behaviors: crossing and jaywalking. We combine the two classes as “crossing” and the rest of the classes in this set as “not crossing”. For the atomic action prediction task, we also choose the atomic action set which includes standing, running, bending, walking, sitting, kneeling, squatting, and lying down. We remove the kneeling, squatting, and lying down since these classes contain too few samples.

**PIE** [27] is a dataset for pedestrian action and trajectory prediction. It records 6 h of 30 FPS driving videos in Toronto with annotations of 30 Hz frequency. The annotations include the binary crossing action labels as well as continuous intentions estimated by human annotators. Since the crossing behavior is only labeled at the end of each track, we select the last 60 frames of all tracks as training and testing samples to align with the setting in TITAN. We also downsample the frequency from 30 Hz to 10 Hz to keep it consistent between the two datasets.

### 4.3. Action Prediction Results

We compare the prediction performance of MulCPred with the following baselines.

(1) *3D CNN* includes C3D [44], R3D18 [49], R3D50 [49], and I3D [50], which are all pre-trained on Kinetics 700 [45]. We feed the appearance modality as input to the 3D CNN models.

(2) *PCPA* [11] is a multi-modal predictor that takes the context, trajectory, ego-motion, and skeleton modalities as inputs. Information from different modalities is fused by a modality attention mechanism.

We compare the crossing prediction performance of the baselines with MulCPred as well as the ablation versions of the regularization loss terms. Table 1 shows the results of 3 classification criteria: accuracy, AUC, and F1 score. *MulCPred* denotes the version with all 5 modalities, while *MulCPred-ASC* denotes the version with only visual modalities, i.e., appearance (A), skeleton (S), and local context (C). Table 2 provides the comparison of different numbers of concepts. We evaluated two extreme cases, i.e., 1 and 20 concepts for each modality, which is equivalent to 5 and 100 concepts in total.

Figure 4 includes the visualization of several concepts. See https://github.com/Equinoxxxxx/MulCPred/blob/main/concepts_visualization.md (accessed on 20 May 2024) for the visualization of all the concepts. MulCPred learns from the following three tasks: crossing prediction on TITAN, atomic action prediction on TITAN (using visual modalities only), and crossing prediction on PIE. Each concept is visualized by three samples from the training set with the highest activation scores corresponding to the concept, i.e., three representative samples in the training set. As shown in Figure 4, each row, from left to right, represents the most representative samples overlaid by the recalibrated heatmap, as well as the relevance scores of the concepts corresponding to the labels. Note that for spatiotemporal modalities (appearance, skeleton, and local context), we only show the frame with the highly activated heatmap. Also, since crossing prediction is a binary classification task, we use only one relevance score for each concept. Figure 5 illustrates the concepts learned in crossing prediction on TITAN without the regularization loss. Based on the quantitative and qualitative results, we have several observations:

(1) The overall performance of MulCPred is competitive compared to the baselines. For the crossing prediction task on TITAN and PIE, the model incorporating all five modalities outperforms the version relying solely on visual modalities. Conversely, for the atomic action prediction task, the model that utilizes only visual modalities shows superior performance. This challenges our initial assumption that atomic actions, like walking and sitting, are less influenced by the pedestrian’s location, whereas the crossing action is significantly related to it.

(2) According to Figure 4, some of the concepts consistently correspond to relevant and recognizable patterns (e.g., crosswalk and lower body), while others have learned patterns that are complex and irrelevant to the prediction for human intuition. For instance, in Figure 4a, concept $\psi_{0}$ captures the pattern of the trajectory on the left side of the view, and $\psi_{40}$ represents the presence of a crosswalk. Conversely, it appears that $\psi_{42}$ is attempting to capture a distant crowd; however, it is unclear why such a pattern would be strongly associated with “not crossing”, as indicated by the relevance score for $\psi_{42}$. Similarly, in Figure 4c, concept $\psi_{1}$ captures the bicycle part, and concept $\psi_{4}$ is capturing the lower body of a pedestrian facing or facing away from the camera, whereas $\psi_{5}$ is seemingly capturing the upper edge of the image. Although the representative samples for $\psi_{5}$ show that the pedestrians are sitting, the aggregator, however, has learned highly positive relevance scores for “run” and “bend” for $\psi_{5}$. In Section 4.4 and Section 4.5, we discuss that these “unrecognizable” concepts, such as $\psi_{42}$ in Figure 4a, $\psi_{25}$ in Figure 4b, and $\psi_{5}$ in Figure 4c, could be a reason for wrong prediction and overfitting.

(3) MulCPred without the regularization terms outperforms the version with the regularization terms, but encounters severe concept collapse. Figure 5 illustrates all 10 concepts of the appearance modality. It can be seen that all the concepts learned the same pattern, and the samples with the highest activation scores are basically the same. The cross-dataset evaluation results in Table 3 also suggest that the model without the regularization terms is less generalizable.

(4) Large coefficients of the regularization terms could decline the prediction performance. It can be seen in Table 1 that, as the value of $\lambda_{1}$ increases, all results gradually drop. In general, the combination of $\lambda_{1}=0.1$ and $\lambda_{2}=0.5$ is considered effective based on the number of optimal outcomes the model achieves. However, while the model without regularization ($\lambda_{1}=0$) shows better performance, it experiences significant concept collapse, as discussed in (3).

(5) The performance does not always improve as the number of concepts increases. In fact, the performance on PIE even decreases considerably as the number of concepts doubles, indicating that only limited features matter to the performance.

(6) Visual modalities have stronger relevance to the predictions. The absolute values of visual modalities, i.e., appearance, skeleton, and local context, are generally larger than the rest. There are two possible reasons for this observation. On the one hand, both crossing and atomic actions can be recognized through visual means rather than by trajectory and ego-motion, especially when the ego-motion is smooth. On the other hand, visual modalities exhibit larger variances compared to the other two, which enhances the saliency that the model can learn.

### 4.4. Failed Cases

We illustrate the inference process of a failed sample in comparison with a correctly predicted sample in Figure 6. Figure 6a shows the case where a person riding a bicycle (labeled as “sitting” in TITAN) is correctly predicted. Among the three concepts we visualized, concept $\psi_{1}$ apparently focuses on the bicycle that the person is riding on, which can also be seen in the recalibrated feature map $\psi_{1}$ applies to the input; $\psi_{14}$ focuses on the lower body part of a sitting person, which also confronts with the recalibrated feature map. Therefore the high activation of both concepts $\psi_{1}$ and $\psi_{14}$ contributes to high confidence in the “sit” class.

On the contrary, Figure 6b shows the case that a person walking laterally to the camera is wrongly predicted as “running”. Since $\psi_{1}$ focuses on the bicycle part, it is reasonable that $\psi_{1}$ has low activation in this sample. However, $\psi_{13}$, which seems to correspond to the lower body part of a person walking laterally to the camera, does not have high activation in this sample either. It is the concept $\psi_{12}$ corresponding to an unrecognizable pattern that matches the input well. Although all pedestrians in the representative samples for $\psi_{12}$ are also walking laterally to the camera, the relevance scores for $\psi_{12}$ learn higher relevance for “run” than “walk”, which contributes to the mistake that the sample is predicted as “run” instead of “walk”.

The comparison between Figure 6a,b shows two typical reasons that could cause failed prediction: (1) **some concepts learned complex patterns** that correspond to multiple classes (such as $\psi_{12}$ in Figure 6), and such concepts are usually unrecognizable from human perspectives; (2) **some of the relevant and recognizable concepts cannot cover all cases as we expect** (such as $\psi_{13}$ in Figure 6b). These observations also indicate that there are gaps between the patterns the model learned and the concepts in human knowledge [51,52]. Filling such gaps is a direction we believe is very important for future improvements.

### 4.5. Cross-Dataset Evaluation

Despite that, some of the concepts do not exactly confront human cognition, as discussed in Section 4.4, the information the concepts provide can still be used in improving performance. By removing unrecognizable and irrelevant concepts, we improved the generalizability of MulCPred. Specifically, we choose a set of concepts that learned recognizable and relevant patterns For the crossing prediction model trained on TITAN, we choose $\psi_{0}$, $\psi_{1}$, $\psi_{8}$, $\psi_{11}$, $\psi_{18}$, $\psi_{20}$, $\psi_{28}$, $\psi_{32}$, $\psi_{40}$, $\psi_{45}$; for the model trained on PIE, we choose $\psi_{0}$, $\psi_{1}$, $\psi_{4}$, $\psi_{7}$, $\psi_{10}$, $\psi_{11}$, $\psi_{12}$, $\psi_{13}$, $\psi_{14}$, $\psi_{15}$, $\psi_{31}$, $\psi_{39}$, $\psi_{42}$. See https://github.com/Equinoxxxxx/MulCPred/blob/main/concepts_visualization.md (accessed on 20 May 2024) for the visualization. (such as $\psi_{1}$ and $\psi_{14}$ in Figure 6), and manually set the relevance scores of the remaining concepts to zero, which is equivalent to removing all the concepts we did not choose from the model. This practice was driven by the simple intuition that, if the features the model extracts from the inputs are explainable, they should also be generalizable. Since both TITAN and PIE have the crossing prediction task, we conducted the cross-dataset evaluation on crossing prediction. The results are shown in Table 3, where MulCPred-filtered denotes the model after the concept removal. “TITAN -> PIE” means the model was trained on TITAN and tested on PIE, while “PIE -> TITAN” means the opposite. It can be seen in the results that the model after the removal outperforms the original model and even baselines with relatively few parameters such as C3D, which is usually considered less likely to encounter overfitting, indicating that the remaining concepts are indeed better generalizable. It is also notable that the improvement brought by the removal is apparent on PIE, a smaller dataset than TITAN, which could indicate that the removed concepts learned from PIE have more severe overfitting problems.

### 4.6. Faithfulness Evaluation

Faithfulness is an important perspective to verify the explainability of a method. It measures how faithfully the importance of features claimed by the model (the relevance scores in our case) reflects these features’ influence on the final prediction [21]. We extend the most relevant first (MoRF) curve [26] to evaluate the faithfulness of MulCPred. MoRF is commonly used as a perturbation-based metric to evaluate the explanations (e.g., heatmaps of input images) of explaining techniques by progressively removing features according to their importance. The formula of MoRF can be recursively described as follows:(8)$$\begin{array}{c} x_{\mathrm{MoRF}}^{k}=\left\{\begin{array}{cc} x & ,\;\mathrm{if}\;k=0\\ g_{\mathrm{rem}}\left(x_{\mathrm{MoRF}}^{k-1},e_{k}\right) & ,\;\mathrm{if}\;k>0 \end{array}\right. \end{array}$$
where x represents the original input, and $g_{\mathrm{rem}}$ is a perturbation function that removes information e (e.g., pixels or features) from x. $E={\left\{e_{k}\right\}}_{k=1}^{L}$ is a sequence of features sorted by their importance (e.g., weights on the heatmaps) in descending order. Let $f_{c}\left(x\right)$ be the output of the model for class *c*. The area under the curve (${\mathrm{AUC}}_{\mathrm{MoRF}}$), composed of all $f_{c}\left(x_{\mathrm{MoRF}}^{k}\right)$, is the quantity of interest. ${\mathrm{AUC}}_{\mathrm{MoRF}}$ evaluates how *faithful* and how *concentrated* the importance claims by the model are. The faster the curve declines, the more faithful the explanations are.

It is natural to apply the MoRF method to evaluate the faithfulness of the weights in the aggregator, but there is one major limitation: the original formula of MoRF does not consider negative relevance. Therefore, we have extended MoRF to accommodate negative relevance by modifying the initial state and the perturbation function, $g_{\mathrm{rem}}$. Specifically, the formula of the extended MoRF is as follows:(9)$$\begin{array}{c} \begin{array}{c} x_{\mathrm{MoRF}}^{k}=\left\{\begin{array}{cc} g_{\mathrm{rem}}\left(x,E^{*}\right) & ,\;\mathrm{if}\;k=0\\ g(x_{\mathrm{MoRF}}^{k-1},e_{k}) & ,\;\mathrm{if}\;k>0 \end{array}\right. \end{array} \end{array}$$
where $E^{*}$ is a subset of E. $E^{*}$ contains all $e_{k}$ with negative importance scores $r_{k}$. *g* is an expanded form of $g_{\mathrm{rem}}$, which can be described as follows:(10)$$\begin{array}{cc} g(x_{\mathrm{MoRF}}^{k},e_{k})= & \left\{\begin{array}{cc} g_{\mathrm{rem}}(x_{\mathrm{MoRF}}^{k-1}, & e_{k}),\;\mathrm{if}\;r_{k}\geq 0\\ g_{\mathrm{rec}}(x_{\mathrm{MoRF}}^{k-1}, & e_{k}),\;\mathrm{if}\;r_{k}<0 \end{array}\right. \end{array}$$
where $g_{\mathrm{rem}}$ is the same perturbation method used in the vanilla MoRF. In our case, $g_{\mathrm{rem}}$ replaces the perturbed features with 0. Whereas $g_{\mathrm{rec}}$ is the inverse version of $g_{\mathrm{rem}}$, which recovers the perturbed features from 0 to the original values. $x_{\mathrm{MoRF}}^{0}$ should maximize $f_{c}\left(x_{\mathrm{MoRF}}^{k}\right)$ if all $r_{k}$ are faithful. The area under the curve of the extended MoRF can be used as the faithfulness metric in cases with negative relevance:(11)$$\begin{array}{c} {\mathrm{AUC}}_{\mathrm{MoRF}}=\frac{1}{L+1}\sum_{k=1}^{L}\frac{f_{c}\left(x_{\mathrm{MoRF}}^{k}\right)+f_{c}\left(x_{\mathrm{MoRF}}^{k-1}\right)}{2} \end{array}$$

We compare the faithfulness of MulCPred and PCPA by *averaging* ${\mathrm{AUC}}_{\mathrm{MoRF}}$ for all samples and all classes in the dataset. We show the ${\mathrm{AUC}}_{\mathrm{MoRF}}$ in Table 4. For PCPA, we evaluate the modality attention values since they function similarly to the relevance scores in MulCPred. MulCPred is better faithful since it learns stable and explicit relevance. The ${\mathrm{AUC}}_{\mathrm{MoRF}}$ for PCPA is relatively high because the modality cannot explicitly represent the relation between the input modalities and different actions to be predicted.

## 5. Discussion

Both quantitative and qualitative results in Section 4 demonstrate that MulCPred is capable of making accurate predictions as well as giving faithful explanations of its inner workings. Although the variant without the regularization terms has better prediction performance, the visualization of the concepts shows that the absence of the regularization terms can cause serious mode collapse. We believe the performance gap caused by the regularization terms indicates that some concepts, though irrelevant and unrecognizable (such as $\psi_{25}$ in Figure 4b and $\psi_{12}$ in Figure 6), can actually cause overfitting to the training dataset. This hypothesis is then demonstrated in the cross-dataset evaluation, where the variant of MulCPred without the regularization has the lowest generalizability. We further find that, by removing these irrelevant and unrecognizable concepts, MulCPred outperforms most baselines, even including C3D, a relatively small model that can be considered less likely to cause overfitting. All in all, despite that there are gaps between the concepts learned by MulCPred and the real concepts in human cognition, MulCPred still shows the potential to give faithful explanations while achieving competitive performance.

## 6. Conclusions

We presented MulCPred, a concept-based model for pedestrian crossing prediction that can explain its own decision-making by learning multi-modal concepts and the relevance between the concepts and the predictions. The model cannot only make sample-level explanations but also attend to different components in the inputs through the recalibration module. In the cross-dataset evaluation, we find that concepts that focus on more consistent and recognizable patterns are more generalizable than the others, which confronts human intuition. Experiment results show that MulCPred can provide competitive prediction performance as well as faithful explanations.

Future work will focus on improving the concepts and generalizing the proposed framework: (1) further disentangling the elements in the context modality to encourage the concepts to learn more consistent and understandable patterns; (2) incorporating the natural language modality to reduce the ambiguity of the explanations; (3) extending the current framework to a wider range of applications, such as trajectory and pose prediction; (4) exploring the possibility of explainable fusion mechanisms more complex than linear combinations for inter-modality fusion, and (5) exploring the potential to apply the proposed framework to arbitrary layers within a neural network.

## Figures and Tables

**Figure 1 sensors-24-06742-f001:**
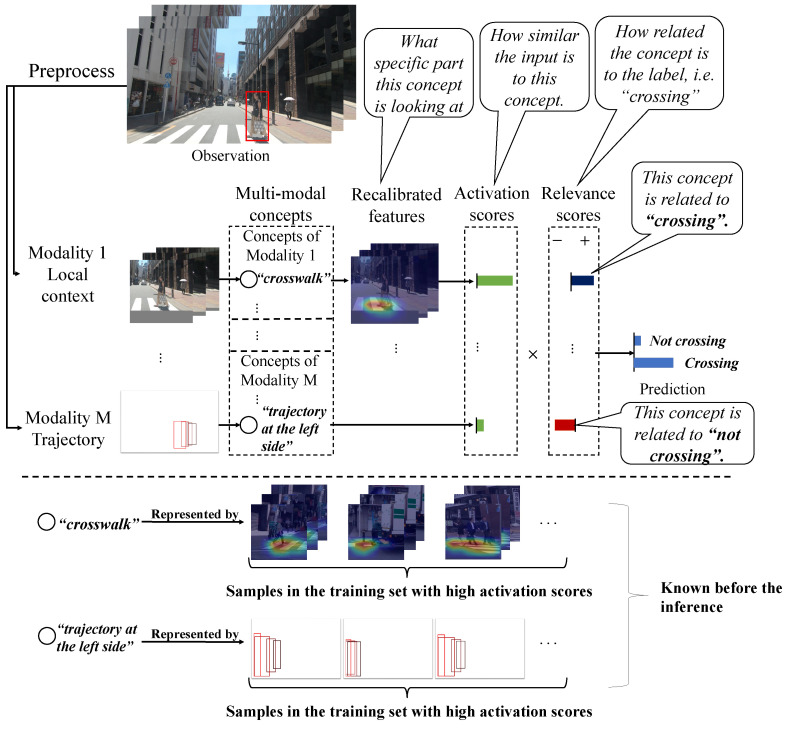
Illustration of MulCPred. The part above the dashed line illustrates the inference process of MulCPred. The activation scores describe the relation between the inputs and the concepts, while the relevance scores describe the relation between the concepts and the prediction. The part below the dashed line illustrates how to explain the concepts.

**Figure 2 sensors-24-06742-f002:**
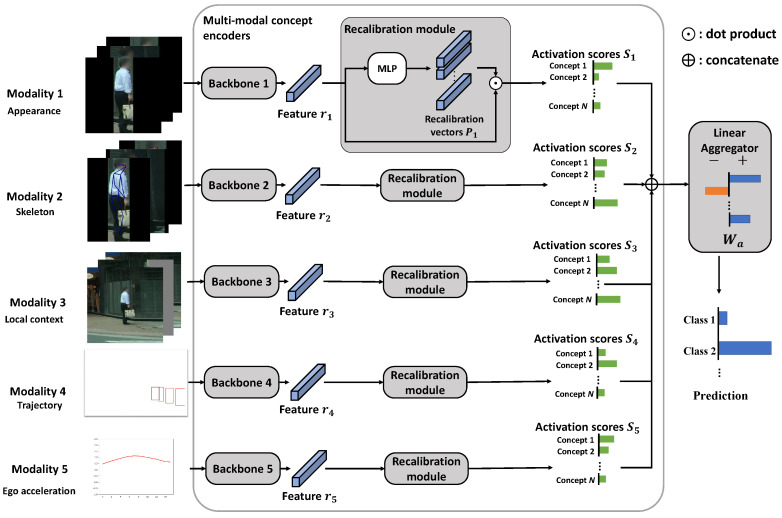
Overall illustration of the MulCPred architecture. Our model takes multi-modal data as inputs, including spatiotemporal data such as videos or sequential data like trajectories. At each modality branch, the input is projected into several activation scores based on its similarity to a set of concepts. Activation scores from all modalities are integrated by a linear aggregator such that the prediction of the model can be explained in an ante hoc manner.

**Figure 3 sensors-24-06742-f003:**
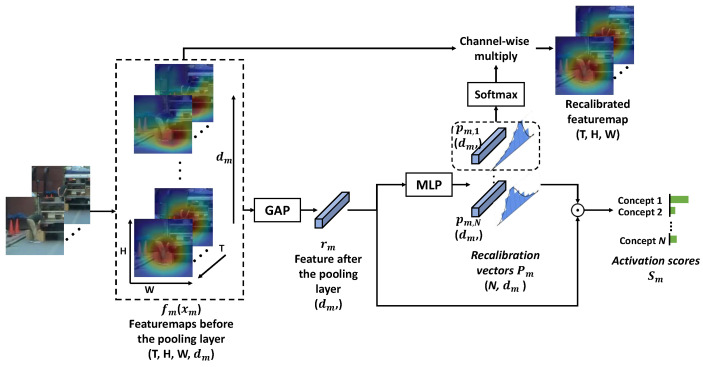
The visualization process of concepts for spatiotemporal modalities. The feature map is first passed to a global average pooling (GAP) layer and then an MLP to be projected into a set of recalibration vectors. Each concept represents a prototype distribution of components in the feature. The heatmap that represents the result of the input activating one concept is obtained by a channel-wise multiplication between the feature and the corresponding recalibration vector.

**Figure 4 sensors-24-06742-f004:**
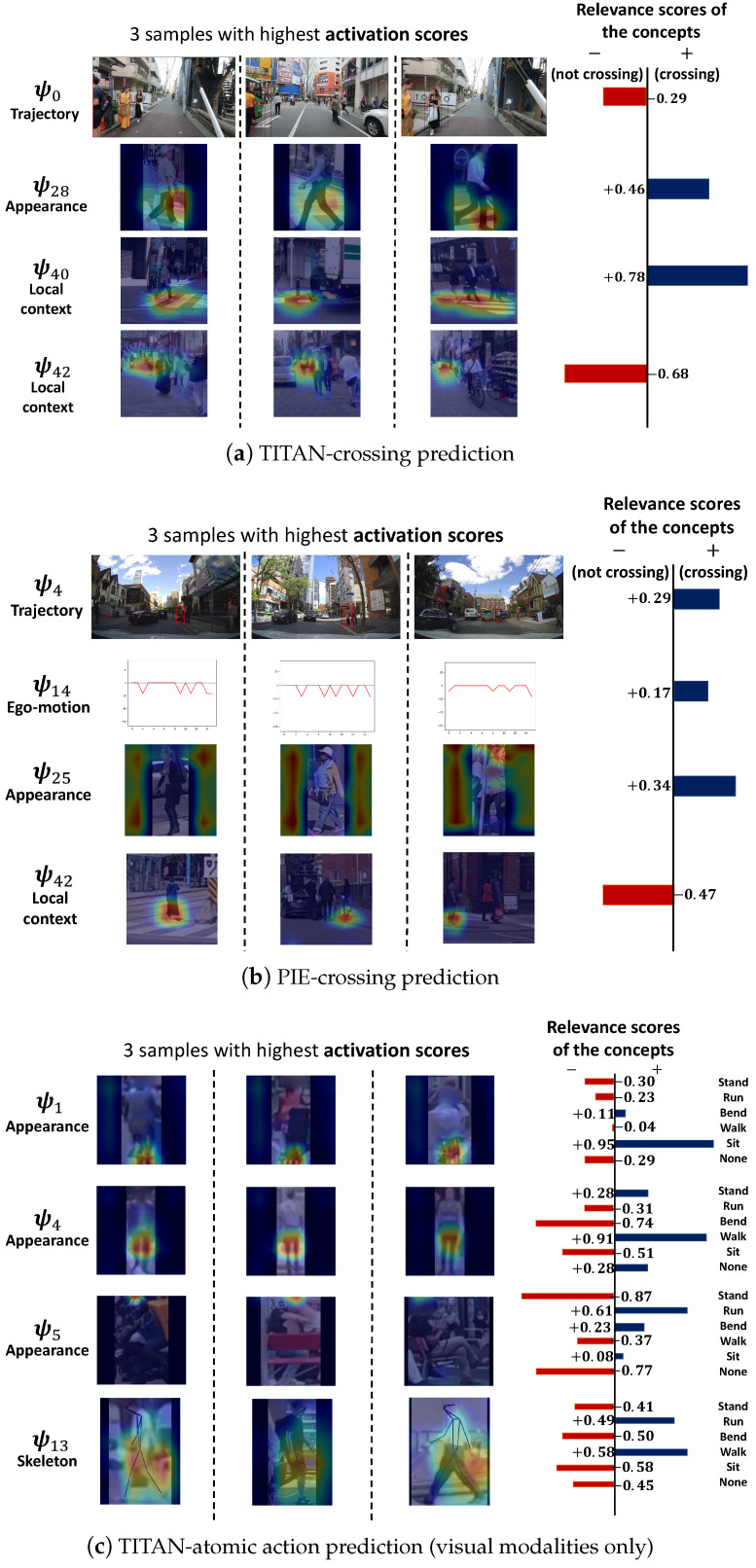
Part of the visualization of the concepts learned from 3 tasks, i.e., crossing prediction in TITAN, crossing prediction in PIE, and atomic action prediction in TITAN. For each concept, we visualize 3 samples in the training set with the highest activation values corresponding to these concepts. It can be seen that some concepts have learned consistent and reasonable patterns such as crosswalk, bicycle, and pedestrian lower body movements, while other concepts have learned patterns that are irrelevant, such as the upper edge of the image and the black padding regions.

**Figure 5 sensors-24-06742-f005:**
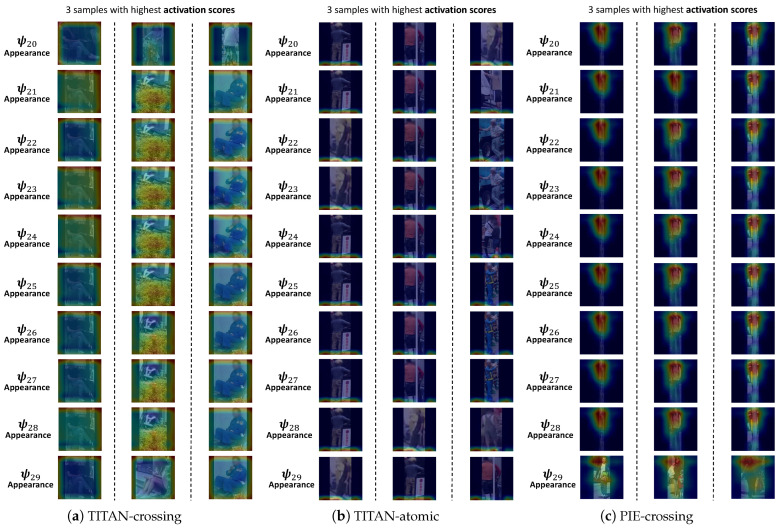
Visualization of all 10 concepts of appearance modality from the TITAN crossing prediction task, TITAN atomic action prediction task, and PIE crossing prediction task when $\lambda_{1}=0$. All these concepts have learned the same pattern, which seems to be the black padding region.

**Figure 6 sensors-24-06742-f006:**
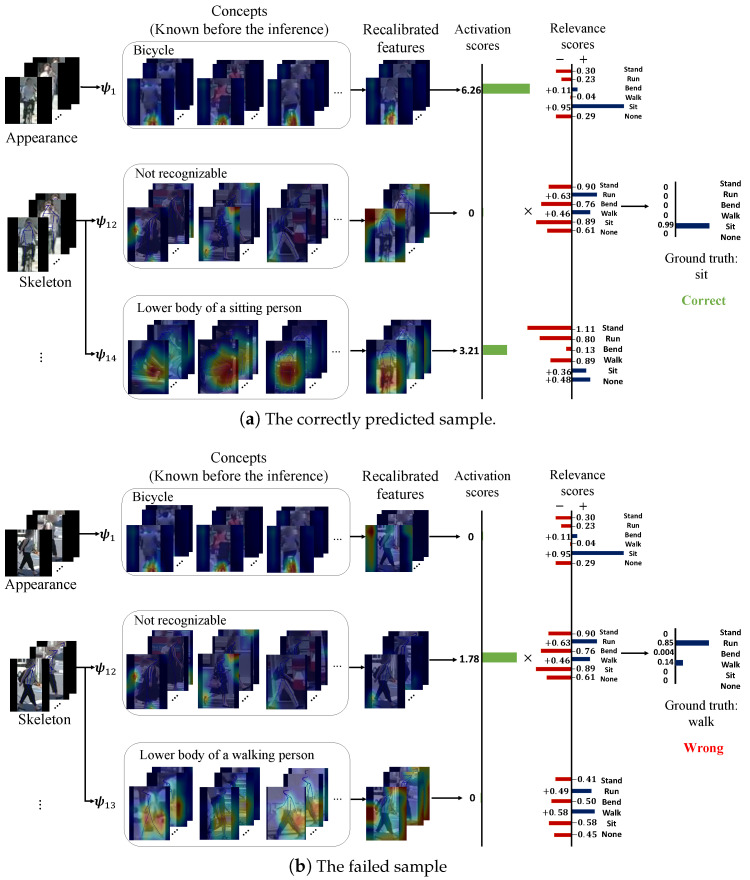
The inference process of a wrongly predicted sample and a correctly predicted sample in the atomic action prediction task on TITAN.

**Table 1 sensors-24-06742-t001:** Action prediction performance.

Methods			TITAN						PIE		
			Crossing Prediction			Atomic Action Prediction			Crossing Prediction		
			Acc↑	AUC↑	F1↑	Acc↑	AUC↑	F1↑	Acc↑	AUC↑	F1↑
C3D			0.87	0.87	0.76	0.86	0.77	0.47	0.87	0.91	0.84
R3D18			0.85	0.89	0.74	**0.90**	0.84	0.52	0.84	0.90	0.80
R3D50			0.89	0.88	0.78	**0.90**	0.84	0.53	0.87	0.91	0.83
I3D			0.88	0.88	0.77	**0.90**	**0.85**	0.52	0.87	0.91	0.83
PCPA			**0.92**	**0.93**	**0.83**	0.86	0.77	0.46	0.88	**0.94**	0.84
MulCPred	$\lambda_{1}=0.1$	$\lambda_{2}=0.5$	0.90	0.90	0.81	0.88	**0.85**	0.52	**0.89**	0.92	**0.86**
		$\lambda_{2}=0$	0.86	0.91	0.76	0.89	0.80	0.51	0.84	0.92	**0.86**
		$\lambda_{2}=1$	0.89	0.88	0.79	0.87	**0.85**	0.51	0.85	0.88	0.79
	$\lambda_{1}=0.5$	$\lambda_{2}=0.5$	0.83	0.88	0.73	0.87	0.82	0.49	0.83	0.87	0.75
	$\lambda_{1}=1$	$\lambda_{2}=0.5$	0.86	0.88	0.76	0.83	0.80	0.43	0.82	0.87	0.79
	$\lambda_{1}=0$	-	0.91	**0.93**	**0.83**	0.88	**0.85**	0.53	**0.89**	0.90	**0.86**
MulCPred-ASC	$\lambda_{1}=0.1$	$\lambda_{2}=0.5$	0.89	0.90	0.80	0.86	0.82	**0.54**	0.84	0.88	0.80

Bold refers to best results in the column. Uparrows mean the higher the better.

**Table 2 sensors-24-06742-t002:** Impact of the number of concepts.

# Concepts	TITAN						PIE		
	Crossing Prediction			Atomic Action Prediction			Crossing Prediction		
	Acc↑	AUC↑	F1↑	Acc↑	AUC↑	F1↑	Acc↑	AUC↑	F1↑
5	**0.90**	0.90	0.80	0.86	0.77	0.47	0.78	0.82	0.72
50	**0.90**	0.90	**0.81**	**0.88**	**0.85**	**0.52**	**0.89**	**0.92**	**0.86**
100	0.88	**0.91**	0.79	**0.88**	0.83	**0.52**	0.68	0.70	0.61

Bold refers to best results in the column. Uparrows mean the higher the better.

**Table 3 sensors-24-06742-t003:** Cross dataset evaluation.

Models		TITAN ->PIE			PIE ->TITAN		
		Crossing Prediction			Crossing Prediction		
		Acc↑	AUC↑	F1↑	Acc↑	AUC↑	F1↑
C3D		**0.59**	0.47	0.39	0.34	0.59	0.34
R3D18		0.56	0.49	**0.44**	**0.57**	0.55	0.47
R3D50		**0.59**	0.53	0.37	0.46	**0.64**	0.43
I3D		**0.59**	**0.64**	0.38	0.54	0.66	**0.48**
PCPA		0.58	0.46	0.37	0.39	0.54	0.37
MulCPred	$\lambda_{1}=0.1$	**0.59**	0.55	0.37	0.35	0.56	0.34
MulCPred	$\lambda_{1}=0$	**0.59**	0.52	0.37	0.32	0.53	0.31
MulCPred-filtered	$\lambda_{1}=0.1$	**0.59**	0.56	0.39	**0.57**	0.50	0.45

Bold refers to best results in the column. Uparrows mean the higher the better.

**Table 4 sensors-24-06742-t004:** Faithfulness comparison in the form of the area under the extended MoRF curve.

Methods	${\mathrm{AUC}}_{\mathrm{MoRF}}↓$		
	TITAN		PIE
	Crossing Prediction	Atomic Action Prediction	Crossing Prediction
PCPA	0.62	0.63	0.65
MulCPred	**0.45**	**0.37**	**0.52**

Bold refers to best results in the column. Downarrows mean the lower the better.

## Data Availability

The data used in this paper are publicly available at https://data.nvision2.eecs.yorku.ca/PIE_dataset/ and https://usa.honda-ri.com/titan (accessed on 31 August 2021).

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
