# Peer review of "MulCPred: Learning Multi-Modal Concepts for Explainable Pedestrian Action Prediction"

_sensors, 2024, doi:10.3390/s24206742_

Round 1

Reviewer 1 Report

Comments and Suggestions for Authors

1. Performance vs. Explainability Trade-off:  While MulCPred aims to improve explainability without significant performance loss, some results show that the performance is slightly lower compared to certain baselines (e.g., PCPA in Table 1).

I suggest provide a more detailed discussion on the trade-off between explainability and performance. Elaborate on scenarios where the slight performance decrease is acceptable due to gains in explainability. Consider quantifying the value of improved explainability in safety-critical applications like autonomous driving.

2. Comparative Analysis with Recent State-of-the-Art Methods: While the paper compares MulCPred with several baselines, it lacks a comparison with the most recent state-of-the-art methods in pedestrian action prediction and explainable AI.

Isuggest expand the related work section to include recent advancements in the field. Update experimental comparisons to include these methods, discussing both performance and explainability aspects. If possible, perform experiments to compare MulCPred with these newer methods.

3.  Terminology Clarification: Terms like “unrecognizable concepts” and “concept collapse” may be unclear to readers unfamiliar with the specific context. I suggestion: Define all specialized terms when first introduced. Provide examples to illustrate these concepts, enhancing the reader's understanding.

4. Regularization and Mode Collapse: The feature regularization loss helps prevent mode collapse but may lead to reduced performance. The best performance is reported when the regularization coefficient (λ1) is zero, which corresponds to no regularization and observed mode collapse (as shown in Figure 5). I suggestion: Investigate alternative regularization techniques that balance the prevention of mode collapse with maintaining or improving performance. Experiment with different values of λ1 and λ2, or consider adaptive regularization strategies. Provide additional analysis on how regularization impacts the diversity of concepts and model generalizability.

5. Discussion on Limitations and Future Work: The paper briefly mentions gaps between model concepts and human cognition but doesn't delve deeply into the limitations of the proposed approach.

Suggestion: Include a dedicated section on limitations, discussing issues such as the subjective nature of concept interpretability, potential biases in concept representations, and challenges in aligning learned concepts with human-understandable ones. Outline future work aimed at addressing these limitations, such as integrating natural language descriptions or incorporating human-in-the-loop training.

Comments on the Quality of English Language

I suggest polishing the English in the manuscript, which will help improve readers' understanding.

Author Response

Reviewer 1

Comment 1:

Performance vs. Explainability Trade-off:  While MulCPred aims to improve explainability without significant performance loss, some results show that the performance is slightly lower compared to certain baselines (e.g., PCPA in Table 1).

I suggest provide a more detailed discussion on the trade-off between explainability and performance. Elaborate on scenarios where the slight performance decrease is acceptable due to gains in explainability. Consider quantifying the value of improved explainability in safety-critical applications like autonomous driving.

Response:

Thank the reviewer for the comment. In fact, Figure 5 provides intuitive examples of the importance of explainability. If inner workings as illustrated in Figure 5 are revealed to the developers or the users, the trustworthiness of the model would be certainly influenced. Considering the above, the loss of performance to some extent is acceptable as the explanations provide promising directions of further improvements and insights of the model, not to mention that the loss is not universal.

As for quantitative comparisons of explainability, we discussed the faithfulness of explanations in Section 4.6 as an important part of explainability. However, the evaluation of the other part, “how well can the explanations be understood”, depends significantly on subjective opinions. Despite we only provide qualitative results in this regard, we admit that it is still an important direction of further improvements.

Supplementary discussion is added to Section 4.3 as follows:

“…Despite the fact that the proposed model did not outperform all baselines in all settings, the slight loss of performance is still acceptable, as the explanations make the model free of doubt of learning ``shortcuts" that human cannot understand and that cannot generalize well. For instance, the proposed model without the regularization terms outperforms the complete version. However, according to Figure \ref{fig5}, the concepts learned by the former collapse to few patterns that are hardly explainable. Considering such a possibility, explainability is certainly worthwhile.”

Thank you for your kind perusal.

Comment 2:

Comparative Analysis with Recent State-of-the-Art Methods: While the paper compares MulCPred with several baselines, it lacks a comparison with the most recent state-of-the-art methods in pedestrian action prediction and explainable AI.

Isuggest expand the related work section to include recent advancements in the field. Update experimental comparisons to include these methods, discussing both performance and explainability aspects. If possible, perform experiments to compare MulCPred with these newer methods.

Response:

Thank the reviewer for the comment. We expanded Section 2 with more recent progress. We’d also like to point out that, although there have been a number of newer methods, many of them still include the practice we discussed in Section 2 that “use separate encoders to generate fixed-length representations of different modalities, and then fuse these representations at a late stage”. Among these methods, the baselines we evaluated are still representative and have competitive performance. Therefore, the results reported are still worth referring to.

Comment 3:

Terminology Clarification: Terms like “unrecognizable concepts” and “concept collapse” may be unclear to readers unfamiliar with the specific context. I suggestion: Define all specialized terms when first introduced. Provide examples to illustrate these concepts, enhancing the reader's understanding.

Response:

We appreciate it that the reviewer pointed this out. Corresponding explanations and examples were added where these terms are discussed (Section 1 and Section 4.3).

Comment 4:

Regularization and Mode Collapse: The feature regularization loss helps prevent mode collapse but may lead to reduced performance. The best performance is reported when the regularization coefficient (λ1) is zero, which corresponds to no regularization and observed mode collapse (as shown in Figure 5). I suggestion: Investigate alternative regularization techniques that balance the prevention of mode collapse with maintaining or improving performance. Experiment with different values of λ1 and λ2, or consider adaptive regularization strategies. Provide additional analysis on how regularization impacts the diversity of concepts and model generalizability.

Response:

Thank the reviewer for the comment. We frame this problem as a part of the trade-off between performance and explainability. As discussed in the manuscript and the comment 1, such trade-off is almost inevitable in practice. By forcing the concepts to be diverse, we actually prevent the model from sacrificing explainability for slight improvements on performance. Despite our concentration on improving explainability, we acknowledge that it is a promising direction of future work to develop techniques that improve the performance without losing explainability.

Supplementary discussion is added in Section 5.

Comment 5:

Discussion on Limitations and Future Work: The paper briefly mentions gaps between model concepts and human cognition but doesn't delve deeply into the limitations of the proposed approach.

Suggestion: Include a dedicated section on limitations, discussing issues such as the subjective nature of concept interpretability, potential biases in concept representations, and challenges in aligning learned concepts with human-understandable ones. Outline future work aimed at addressing these limitations, such as integrating natural language descriptions or incorporating human-in-the-loop training.

Response:

Thank the reviewer for the comment. We have added a focused discussion in Section 5 about the limitations of the method as follows:

“The reported results revealed the following limitations of the proposed method:

1) The gap between the concepts the model learns and the ones human understand. As shown in Figure \ref{fig6}, certain concepts learn patterns such as edges of the images, padding regions, etc, which we refer to as ``unrecognizable concepts", though certain measures such as regularization terms are applied. This phenomenon indicates that there is still a gap between the concepts learned by the model and the ones in human knowledge, and that further techniques can be developed to bridge the gap, including but not limited to more effective regularization and more high quality data.

2) Lack of a comprehensive metric for explainability. Although we argue that faithfulness is an essential part of explainability, it cannot quantify the extent to which human would understand the explanations. To overcome the inherent subjectivity, conclusions from interdisciplinary study such as cognitive science can be considered as references.”

Thank you for your kind perusal.

Reviewer 2 Report

Comments and Suggestions for Authors

I am writing this review from a mobility specialist’s perspective. This manuscript has improved on current models of pedestrian action prediction for two tasks: pedestrian crossing prediction and pedestrian atomic action condition by using five different input modalities: appearance, skeleton, local context, trajectory, and ego-motion. All of these are important to improve learning of the model. The authors demonstrated the different weights of these models on improving pedestrian action prediction in a variety of contexts. They provided a number of different scenes and situations to demonstrate the effectiveness of this model. I have no major issues with the manuscript’s publication as it is written. I have only a few comments and questions for the authors to consider:

Considering that these models are based on the TITAN and PIE datasets which are from Tokyo and Toronto, respectively, I see a value of providing additional datasets for the learning model to improve learning and potentially better understand the difference in context in different regions of the world. For instance, are there certain cultural differences related to crossing behaviors, such as for someone in India or Nigeria compared to these contexts? How predictive is this model in that context. Also, when there is snow or ice on the road, how do these models perform, due to potential contrast and reflective differences? Also, crosswalks vary considerably in different regions of the world in terms of color, design, shape and height. It appears as if this model could benefit from learning different crosswalks in various locations around the world.

With the skeleton modality, does it make an accurate prediction of the age of the individual, because the crossing behavior of a child, young adult or older adult may differ. For instance, a young adult may be more likely to run in front of the car. Does the skeleton modality examine gait patterns, which also could be indicative of age or crossing behavior?

Overall, what is the speed of accurate prediction with this model? It is feasible to have an accurate prediction in a timely manner if the vehicle is traveling over 50 or 70 km/h?

It appears that direct human input could enhance the predictability of the model even further by learning potential differences in travel with various mobility devices (walker, wheelchair, scooters, electric scooters, skateboards, etc.). Especially considering that active mobility devices are in constant evolution.

Author Response

Reviewer 2

Comment 1:

Considering that these models are based on the TITAN and PIE datasets which are from Tokyo and Toronto, respectively, I see a value of providing additional datasets for the learning model to improve learning and potentially better understand the difference in context in different regions of the world. For instance, are there certain cultural differences related to crossing behaviors, such as for someone in India or Nigeria compared to these contexts? How predictive is this model in that context. Also, when there is snow or ice on the road, how do these models perform, due to potential contrast and reflective differences? Also, crosswalks vary considerably in different regions of the world in terms of color, design, shape and height. It appears as if this model could benefit from learning different crosswalks in various locations around the world.

Response:

We thank the reviewer for the comment. Due to the space limitations, we cannot evaluate all possibilities. We hope the following explanations could be preliminary answers to your questions.

The domain gap of different environments is indeed a worth studying problem. One practical reason we only use two datasets is that public datasets with precise manual labels of human actions are limited. Also, we agree that a universal model that can adapt to all existing rules and environments is worth looking forward to, but the necessity of doing so depends on the scenarios that the agent actually encounters in real applications. If the model is only applied to limited scenarios, incorporating training data from multiple domains might not necessarily improve the performance, but from the perspective of explainability, I believe it is helpful for the model to learn more abstract and more fundamental features.

Comment 2:

With the skeleton modality, does it make an accurate prediction of the age of the individual, because the crossing behavior of a child, young adult or older adult may differ. For instance, a young adult may be more likely to run in front of the car. Does the skeleton modality examine gait patterns, which also could be indicative of age or crossing behavior?

It appears that direct human input could enhance the predictability of the model even further by learning potential differences in travel with various mobility devices (walker, wheelchair, scooters, electric scooters, skateboards, etc.). Especially considering that active mobility devices are in constant evolution.

Response:

Thank the reviewer for the comment. We believe that all these practices of adding additional inputs such as age and device, can help improve the model, especially when the training data are limited, since they implicitly provide the knowledge structure human developed to reason about the world. However, as suggested in the comment, some features such as age are originated from the skeletons or appearance. As the amount of training data increases, it is likely that the model could learn to derive these features automatically from raw inputs. All in all, we do believe that more diverse training data can bring interesting results.

Supplementary discussion is added to Section 6:

“… and 6) exploring the utility of the rich data without action labels.”

Comment 3:

Overall, what is the speed of accurate prediction with this model? It is feasible to have an accurate prediction in a timely manner if the vehicle is traveling over 50 or 70 km/h?

Response:

Thank the reviewer for the comment. The precise running time of the model depends on multiple factors, including hardwares, frameworks of deployment, the versions of packages, and most importantly, the backbones used as feature encoders, since they take the majority of the size of the model. Therefore, we’d rather provide the parameters that are constant in the model, e.g. the number of dimensions and concepts, and open the code for arbitrary settings.

Reviewer 3 Report

Comments and Suggestions for Authors

The manuscript presents an innovative approach to pedestrian action prediction by introducing the MulCPred framework, which leverages multi-modal concepts for explainability. The proposed method addresses notable limitations in concept-based prediction models, such as the inability to handle multi-modal data, lack of locality in predictions, and mode collapse. The novelty of the work lies in its use of a linear aggregator combined with a channel-wise recalibration module to create more transparent and interpretable predictions. These contributions are well-founded and advance the state-of-the-art, explainable AI for pedestrian action prediction.

The work is significant for several reasons. First, it targets a critical aspect of autonomous driving—predicting pedestrian actions with improved interpretability. The model enhances trustworthiness by providing explanations for predictions, which is essential for safety-critical applications. Additionally, the framework's ability to generalize across different datasets, as demonstrated by the cross-dataset evaluations, suggests practical utility in real-world scenarios. The authors’ exploration of regularization techniques to combat mode collapse and improve generalizability further highlights the depth and relevance of their contributions.

The manuscript is generally well-organized and clearly presented. The theoretical foundations and the structure of the MulCPred model are comprehensively explained. The visual aids, including figures and tables, effectively support the text, illustrating key concepts and experimental results. However, certain sections, such as the technical details of the linear aggregator and the feature recalibration module, could benefit from further clarification to make them more accessible to readers who may not be deeply familiar with these methodologies.

The scientific rigour of the work is commendable. The authors conduct extensive experiments on multiple datasets, employing quantitative and qualitative metrics to validate their claims. The methodology is sound, and the comparisons with baseline models are thorough and appropriate. Nonetheless, there are a few areas where additional analysis could be beneficial, such as a deeper exploration of the impact of different regularization techniques on model performance. Furthermore, more insight into the proposed model's limitations and potential biases would enhance the conclusions' robustness.

Given the increasing focus on the explainability of AI models, particularly in safety-critical applications like autonomous driving, this work will likely attract significant interest from the research community. The combination of pedestrian action prediction and explainable AI is a timely and relevant topic. Moreover, the paper’s emphasis on multi-modal data fusion and its implications for model interpretability make it an essential contribution to fields such as computer vision and machine learning.

Overall, this manuscript makes a valuable contribution to the field of explainable AI and pedestrian action prediction. The novel methodological advancements and a robust evaluation framework establish MulCPred as a promising tool for research and practical applications. While there is room for improvement in the presentation and additional analysis, the current work is of high quality. It is likely to be of considerable interest to researchers and practitioners alike.

In the future with this work consider adding additional experiments or analyses to explore the effects of different regularization strategies on model performance. Include a more detailed discussion of the model’s limitations, potential biases, and avenues for future research.

Author Response

Reviewer 3

Comment 1:

In the future with this work consider adding additional experiments or analyses to explore the effects of different regularization strategies on model performance. Include a more detailed discussion of the model’s limitations, potential biases, and avenues for future research.

Response:

Thank the reviewer for the comment. We have discussed the possibility of more effective regularization in Section 4.3, and also dedicated a part of discussion on limitations of the model. The supplementary discussion is as follows:

“The reported results revealed the following limitations of the proposed method:

1) The gap between the concepts the model learns and the ones human understand. As shown in Figure \ref{fig6}, certain concepts learn patterns such as edges of the images, padding regions, etc, which we refer to as ``unrecognizable concepts", though certain measures such as regularization terms are applied. This phenomenon indicates that there is still a gap between the concepts learned by the model and the ones in human knowledge, and that further techniques can be developed to bridge the gap, including but not limited to more effective regularization and more high quality data.

2) Lack of a comprehensive metric for explainability. Although we argue that faithfulness is an essential part of explainability, it cannot quantify the extent to which human would understand the explanations. To overcome the inherent subjectivity, conclusions from interdisciplinary study such as cognitive science can be considered as references.”

Thank you for your kind perusal.

Round 2

Reviewer 1 Report

Comments and Suggestions for Authors

I suggest polishing the English in the paper, which can enhance its readability.

Comments on the Quality of English Language

 Minor editing of English language required.

Author Response

Comments: I suggest polishing the English in the paper, which can enhance its readability.

Response: We thank the reviewer for the comments. We had the English-speaking authors do the proofreading and polish the language. We also corrected several errors in the expressions. Detailed modifications are highlighted in the supplementary materials.